# Reporting of 3Rs Approaches in Preclinical Animal Experimental Studies—A Nationwide Study

**DOI:** 10.3390/ani13193005

**Published:** 2023-09-23

**Authors:** Birgitte S. Kousholt, Kirstine F. Præstegaard, Jennifer C. Stone, Anders F. Thomsen, Thea T. Johansen, Merel Ritskes-Hoitinga, Gregers Wegener

**Affiliations:** 1Department of Clinical Medicine, AUGUST, Aarhus University, 8200 Aarhus, Denmark; kfp@clin.au.dk (K.F.P.); afth@clin.au.dk (A.F.T.); tjo@biomed.au.dk (T.T.J.); j.ritskes-hoitinga@uu.nl (M.R.-H.); wegener@clin.au.dk (G.W.); 2JBI, Faculty of Health and Medical Sciences, The University of Adelaide, Adelaide, SA 5005, Australia; j.stone@adelaide.edu.au; 3Department of Health Services Research and Policy, Research School of Population Health, Australian National University, Canberra, ACT 2600, Australia; 4Institute for Risk Assessment Sciences, Faculty of Veterinary Medicine, Utrecht University, 3584 CM Utrecht, The Netherlands; 5Translational Neuropsychiatry Unit, Department of Clinical Medicine, Aarhus University, 8200 Aarhus, Denmark

**Keywords:** 3Rs, reporting, preclinical animal studies, refinement, reduction, replacement, ARRIVE, animal welfare

## Abstract

**Simple Summary:**

In society, there is a solid demand to limit the number of animals used in research, improve the animals’ living conditions, and find alternatives to animal experiments. “The 3Rs” is a concept that aims to reduce, refine, and replace animal experiments. It is embedded in legislation worldwide. Even so, finding 3Rs ideas relevant to one’s own research area may be difficult. Recent studies showed that 3Rs approaches from fellow researchers are a popular way to obtain ideas. In this nationwide study, we investigated if 3Rs approaches were reported and described in primary preclinical animal experimental literature. If so, this may be an easy route to access readily available 3Rs ideas and comply with legislation, public demand, and ethical obligations. Our results, however, show that minimal 3Rs information is reported in these studies, which had at least one author affiliated to a Danish university. The level of information varies and is insufficient to learn and implement relevant 3Rs approaches in one’s research. Instead, 3R-specific education and research are needed, and funders should support such research.

**Abstract:**

The 3Rs aim to refine animal welfare, reduce animal numbers, and replace animal experiments. Investigations disclose that researchers are positive towards 3Rs recommendations from peers. Communication of 3Rs approaches via primary preclinical animal experimental literature may become a fast-forward extension to learn relevant 3Rs approaches if such are reported. This study investigates 3Rs-reporting in peer-reviewed preclinical animal research with at least one author affiliated to a Danish university. Using a systematic search and random sampling, we included 500 studies from 2009 and 2018. Reporting was low and improvement over time limited. A word search for 3R retrieved zero results in 2009 and 3.2% in 2018. Reporting on 3Rs-related sentences increased from 6.4% in 2009 to 18.4% in 2018, “reduction” increased from 2.4% to 8.0%, and “refinement” from 5.2% to 14.4%. Replacement was not reported. Reporting of the methodology was missing. For “reduction”, methodology was mentioned in one study in 2009 and 11 studies in 2018, and for “refinement” in 9 and 21, respectively. Twenty-one studies stated compliance with ARRIVE-guidelines or similar without disclosure of details. Reporting of 3Rs approaches in preclinical publications is currently insufficient to guide researchers. Other strategies, e.g., education, interdisciplinary collaboration, and 3Rs funding initiatives, are needed.

## 1. Introduction

The 3Rs principles of reduction, refinement, and replacement gathered speed in 1959 with Russel and Burch’s book “The principles of humane experimental technique” [1]. The concept originates from the scientific community to guide researchers on the ethical use of animals [2]. The concept has evolved, and Norecopa—the Norwegian national consensus platform for the advancement of the 3Rs in connection with animal experiments—summarizes these principles in the following manner: “Replacement alternatives are methods which permit a given purpose to be achieved without conducting procedures on animals; Reduction alternatives are methods for obtaining comparable levels of information from the use of fewer animals in scientific procedures, or for obtaining more information from the same number of animals and Refinement alternatives are methods which alleviate or minimize potential pain, suffering or distress, and which enhance animal well-being and the quality of research…” [3].

The principles are embedded in the European Directive 2010/63 on protecting animals used for scientific purposes and, thus, in European countries’ legislation on animal experimentation. Hence, the 3Rs must be disseminated across the scientific community. Scientists who perform animal experiments must comply with the 3Rs, and research institutions are obligated to advise on the 3Rs via their animal welfare bodies [4]. There is also a rise in society to actively incorporate the 3Rs in animal experiments in the quest towards phasing out animal experiments per se. Recently, an EU citizen petition to ban animal experiments for cosmetic ingredient testing was signed by more than 1.4 million EU citizens [5]. In addition to the legal requirements and the public demand, implementing the 3Rs is essential to bring down the “signal to noise” variation of scientific outcomes. For example, minimizing unwanted stress lowers variation in scientific outcome results and may be addressed through refinement [6,7,8]. Several initiatives and guidelines endorse the 3Rs, e.g., the PREPARE guideline on planning animal experiments [9] and the ARRIVE guideline on reporting animal experiments [10]. The aforementioned raises the question of how well the 3Rs are indeed implemented. A few investigations examined the matter, and one of the main obstacles in the 3Rs’ implementation seems to be difficulties in easy access to 3Rs ideas that researchers can implement in their experiments. An investigation by van Luijk et al. showed that scientists generally lack time to investigate and comply with the 3Rs and need help finding relevant databases to seek inspiration [11]. Another study showed that researchers are concerned with ethical issues and discuss these with their peers [12]. Nevertheless, there still seems to be less focus on the 3Rs when sharing information on, e.g., animal models used in drug discovery [13].

Interestingly, most scientists were attentive to learning about relevant 3Rs approaches from their peers, whereas searches in dedicated 3Rs databases were considered cumbersome [11]. The latter was speculated to be due to a lack of knowledge in searching and obtaining the relevant 3Rs information. Machine learning skills may be helpful in the pursuit to speed up the process and identify relevant 3R initiatives [14]. Other ways to access 3Rs ideas should be scrutinized as well. Across a scientific field, a helping hand may come from peers who convey 3Rs approaches instituted in their experimental animal studies and inform on these in their primary preclinical animal experimental publications. To our knowledge, it is not disclosed to which extent preclinical animal experimental publications mention and promote 3R approaches. However, systematic reviews of preclinical animal experimental studies within specific science fields have shown that there is a gap when evaluating 3Rs practices [13,15]. Nevertheless, such a straightforward information route would provide researchers with 3Rs information relevant to their own field of science. This information would then be available in the literature searches that researchers already perform.

In this nationwide study, we aim to assess the extent to which 3Rs approaches are reported in peer-reviewed preclinical animal experimental studies. To determine if there is a trend toward better disclosure, we conducted this investigation in publications published before the ARRIVE guidelines and the formation of the EU directive 2010/63 (2009) and after (2018). We additionally conveyed the overall reporting status of 3Rs approaches with a focus on the detail level given for each reported item.

## 2. Materials and Methods

### 2.1. Study Protocol

The study protocol is part of a more extensive investigation and is found in the Appendix A [1,9,15,16,17,18]).

### 2.2. Data Sources and Eligibility Criteria

The literature search, random sampling, and data retrieval were described previously [19]. Briefly, the literature search was conducted in Medline (via PubMed) and Embase for all citations that referred to in vivo studies conducted on non-human vertebrates with one or more authors affiliated with at least one of five Danish universities of interest. The search was divided into two separate searches based on publication year (search 1: 1st of January 2018 until 6th of November 2018; search 2: the year 2009) to evaluate the change in reporting of 3Rs approaches over time (the year 2009 before ARRIVE guidelines and EU directive 2010/63, and the year 2018). After removing duplicate citations, 1161 citations from 2009 and 1890 citations from 2018 were retrieved. Studies were randomly sampled to reduce the potential for bias using the “=RAND()” command in Microsoft Excel (Excel, MS Office, version 2016, Microsoft Corp., Redmond, WA, USA) to allocate a unique random number to each publication. Publications were imported consecutively in randomized order into a systematic review manager software program, Covidence (Covidence, Melbourne, Australia) [20]. A total of 1800 studies (out of 3051) were screened for eligibility. Exclusion of studies was based on these criteria: science related to wild animals, farming, invertebrates, human (clinical) studies, environment, in vitro research, not primary publications, lack of abstract or full text, and exploratory studies. The exploratory studies were identified through either study author statements describing that the study was explorative, studies that were assessed to investigate novel questions and to be hypothesis-generating, and/or studies containing no intervention. All studies had at least one author affiliated to a Danish university. Two hundred and fifty-six studies from 2009 and 275 from 2018 were found eligible. We estimated that a sample size of 250 from each year would be feasible to review in detail. These 250 publications were selected based on the random sampling allocation sequence. The PRISMA flow diagram is available in Kousholt et al. [19].

### 2.3. Data Extraction and Analysis

The Covidence Risk of Bias tool was used for this study. It is normally used to assess risk of bias and extract scientific data from studies included in a systematic review and meta-analysis. It is possible to add items or topics of interest relevant to the specific systematic review protocol. Here, the software was selectively modified for the aim of this study so that relevant items concerning the 3Rs were added, e.g., “reduction”, “refinement”, and “replacement”. Each publication was assessed for reporting a search word for “the 3Rs” and for reporting 3R-related sentences indicating reduction, refinement, and replacement in the study setup. For comparison to 3Rs reporting, publications were evaluated according to consideration of compliance with legislation on animal experimentation. Furthermore, reporting of the 3Rs separately and compliance with the 3Rs or ARRIVE guidelines were investigated. Details such as methods for implementation of replacement (e.g., non-animal methods), reduction (e.g., re-use methods), and refinement (housing aspects, e.g., increased cage size, social housing, enrichment of cage environment and food; procedural aspects, e.g., the use of anesthesia, analgesia, humane endpoints, and training for procedures with positive reinforcement) were additionally evaluated. Two independent reviewers (K.F.P. and J.C.S.) assessed publications for compliance with legislation on animal experimentation and the 3Rs approach, each blinded to the other’s assessment. Reviewers examined the full text of the articles, including figures and tables, in addition to Appendix A; references to other studies were not evaluated. The approach included three steps:

Step 1: To investigate the overall reporting status of the selected items, each item was operationalized and scored “Yes” or “No” in Covidence. Briefly, publications were qualitatively scored “Yes” if the specific item was reported and “No” when there was no reporting of the item. “Unclear” or “Partial” scores were not used. In instances where the item was only partially reported and did not contain the complete information stated by the item, the study was scored as “Yes” and notes provided (e.g., reporting of sentences inferring that one or more of the 3Rs were implemented but with no specific implementation measure). Annotations and quotes for each item were selected and saved for subsequent data quantification. This extra step made judgment decisions during this review consistent. Details of this 3-step approach and examples of quotes are provided in the Appendix A.

Step 2: After completing the initial assessments by each reviewer, a consensus of the results was undertaken in Covidence. If both reviewers agreed on the item, the final judgment defaulted to the agreed value leaving discrepant items for further assessment. Discrepancies were resolved and consensus was reached through discussion and the inclusion of a third reviewer (B.S.K.).

Step 3: Data were extracted and sorted in MS Excel. After that, a numerical score of 1, 2, 3, or 0—where 0 corresponds to no information—was given according to the quality of information (quotes and comments) saved for each item described in Step 1.

Survey data were analyzed using MS Excel and Stata Statistical Software: Release 16.1 (Stata Corp. 2019. College Station, TX, USA: StataCorp LLC). Descriptive statistics were generated for all items and presented in bar graphs and tables. Prevalence was reported with 95% confidence intervals. Differences in prevalence between the 2009 and 2018 studies were reported with approximate 95% confidence intervals.

## 3. Results

We assessed publications for reporting on the 3Rs and any measures taken to implement these for the ethical use of animals in experimental research. We also investigated reporting of compliance with legislation on animal experimentation. Five hundred publications were included in this investigation: 250 from 2009 and 250 from 2018.

Publications reporting on the 3Rs identified by a word search increased from zero in 2009 to eight publications (3.2%, 95% CI [1.4–6.2%]) in 2018. Publications reporting sentences indicating reduction, refinement, and replacement increased from 16 publications (6.4%, 95% CI [3.7–10.2%]) in 2009 to 46 publications (18.4%, 95% CI [13.8–23.8%]) in 2018. Compliance with legislation on animal experimentation was reported in 219 publications (87.6%, 95% CI [82.9–91.4%]) in 2009, and this number increased to 242 publications (96.8%, 95% CI [93.8–98.6%]) in 2018 (Figure 1).

When assessing each of the 3Rs separately, we found an increase in reporting on reduction from six publications (2.4%, 95% [CI 0.9–5.2]) in 2009 to 20 publications (8.0%, 95% CI [5.0–12.1%]) in 2018. Publications reporting on refinement increased from 13 publications (5.2%, 95% CI [2.8–8.7%]) in 2009 to 36 publications (14.4%, 95% CI [10.3–19.4%]) in 2018. None of the studies from each research year reported on replacement. Twenty-one publications (8.4%, 95% CI [5.3–12.6%]) from 2018 reported the study to be conducted in compliance with the 3Rs principles (1.2%, 95% CI [0.3–3.5%]) or the ARRIVE guidelines (7.2%, 95% CI [4.3–11.1%]) (Figure 2).

Of the publications reporting on reduction in 2009, one publication (0.4%, 95% CI [0.0–2.2%]) disclosed the method engaged (using animals from a previous study to reduce the animal number). This number increased to 11 publications (4.4%, 95% CI [2.2–7.7%]) in 2018 (reporting statistical power calculation, reusing animals, and improving study design (e.g., combining assays) to reduce the animal number). Of the publications from 2009 that described refinement strategies, nine (3.6%, 95% CI [1.7–6.7%]) disclosed the method used (e.g., reporting a description of humane endpoints, habituating animals to reduce stress, and improved environmental enrichment). This number increased to 21 publications (8.4%, 95% CI [5.3–12.6%]) in 2018. Three publications (1.2%, 95% CI 0.3–3.5%) from 2018 reporting compliance with the ARRIVE guidelines included a checklist (Table 1). The three checklists were one ARRIVE guideline checklist, one word document checklist of ARRIVE items 5–13 (ARRIVE methods), and one reporting checklist—including a recommendation on consulting the ARRIVE guideline and confirming compliance with ARRIVE—provided by the publishing journal.

## 4. Discussion

Surveys investigating the knowledge of and experiences with the 3Rs among researchers involved in animal experiments have identified various obstacles to 3Rs’ implementation [11,21,22,23]. In this study, we surveyed whether researchers disclose 3Rs approaches in their primary publications. We looked into this partly because researchers in a previous investigation mentioned that they obtained a high level of 3R information through the network within their own field of research. Even though this communication was mainly from person to person, a fast-forward way to accelerate communication would be if researchers could retrieve applicable 3Rs approaches when reading already relevant scientific papers. Knowledge and suggestions from peers regarding the 3Rs is generally well-received and appreciated by researchers [11].

### 4.1. Reporting of the 3Rs

Our data show a low reporting of the 3Rs in preclinical animal experimental studies with little improvement over time. We found limited information when investigating the publications for the 3Rs per se (word search for the concept 3R) in both research years. However, a notable increase from 2009 till 2018 in the reporting of 3R-related sentences points towards an increased awareness in the scientific community. This trend aligns with the growing awareness of the 3Rs [24].

### 4.2. Reporting of Each of the Three Rs

Interestingly, when separating the 3Rs into replacement, reduction, and refinement items, we found a discrepancy in how well these items were reported. Reporting of refinement and reduction was present, although limited, whereas replacement was not reported at all. This finding aligns with a report from Nøhr et al., who investigated 3Rs experiences among Danish researchers involved in animal experiments. They found that most researchers consider refinement and reduction approaches when they plan and carry out experiments, whereas replacement is seldom considered [22]. We speculate that replacement is not mentioned at all in the publications included in our study because it is not common practice to emphasize replacement approaches in animal experimental publications. Once a decision to perform an animal experiment is taken, the efforts to replace animals may be considered of less relevance to report. In addition, several studies that have considered and found replacement alternatives are published in dedicated publications disclosing in vitro methods and results. Finally, researchers that use mainly non-animal alternatives may often not see their methods as an alternative to animal experiments. Altogether, this renders it difficult to search and find these replacement alternatives. It may be worthwhile to discuss the 3Rs and raise awareness among researchers occupied with replacement techniques to establish interdisciplinary collaborations and aim towards the disclosure of methods as alternatives to animal experimental studies. This would serve as an additional means to increase the transparency of replacement opportunities.

### 4.3. Reporting of 3Rs Methodological Details

For 3Rs approaches to be easy to implement, thorough information on “how to” is needed. Hence, we further researched the level of detail in the 3Rs information disclosed. In terms of reporting refinement and reduction implemented in one’s study, only a few scientists reported relevant details. The incomplete reporting of these details directly impedes translating information into practice. Researchers who wish to apply, e.g., a reported refinement initiative such as improved housing conditions need detailed information on how this is achieved. Thus, obtaining sufficient information via primary preclinical animal experimental literature on 3Rs approaches is not yet feasible. The question is if it is possible to institute incentives making it attractive to researchers to work with and disclose 3Rs approaches in detail. Financial incentives as a mean to change health behaviors was, e.g., recently discussed by Vlaev et al. [25]. At Aarhus University, local forces with roots in the AUGUST group (a consortium striving to facilitate the understanding of systematic reviews and meta-analyses and work toward improved implementation of the 3Rs principles in science), in 2019, instituted a 3Rs prize via the animal welfare body at the faculty of medical sciences. The prize entailed an economic incentive, namely, EUR 1340, to a Ph.D. student for making extra efforts to implement the 3Rs in their Ph.D. project. The interest for the 3Rs prize has gained traction over the years, implying that it is an effective and positive way to create 3Rs awareness and compliance [26,27]. Another stricter way to introduce change may be if publishers request this reporting more vigorously. Notably, compliance with legislation on animal experimentation was reported on a large scale in both research years, approximating 100% in 2018. This underpins that whenever journals enforce reporting by default, researchers comply. One may argue that since the 3Rs are embedded in most legislation, reporting of details regarding 3Rs approaches should be enforced by default as well. Interestingly, several publishers discourage lengthy methods sections. This push to limit manuscript length may also lead to leaving out important details on the 3Rs. To satisfy everyone, an idea could be to upload a particular 3Rs Appendix A that discloses all 3Rs approaches in detail.

### 4.4. Reporting of ARRIVE Compliance

The ARRIVE guidelines on reporting of animal experiments endorse the 3Rs. The first edition of the guidelines was published in 2010 [16], and we did not expect any reporting considering this item in 2009. Only 18 publications out of 250 studies from 2018 were reported to comply with ARRIVE. When we assessed these publications in detail, no manuscript approached full compliance. This is quite disappointing. Our findings agree with prior reports suggesting that solely endorsing the ARRIVE guidelines is insufficient to obtain compliance [17,28,29,30]. Recent findings suggest that mandated completion of a reporting checklist in ten Nature Journals improved the reporting quality compared to comparator journals [31]. In our study, three publications included a guideline checklist. Our results imply that submitting a guideline checklist will not improve full compliance with the guidelines. We found inconsistencies between what authors claimed on the submitted checklist and what was reported in the published paper, e.g., important 3R-related information such as welfare-related assessments and the health status of experimental animals was reported as “not relevant” or lacking. Reference to refinement approaches was provided in two of the three checklists. The statements indicated what we can only interpret as either very little time spent to fill out information or little knowledge regarding refinement and animal welfare.

### 4.5. Fast-Forwarding 3Rs Implementation—A Shared Responsibility

Our data emphasize the need within the scientific community for discussions on how and where 3Rs approaches should best be described and reported. In 2011, van Luijk et al. reported that scientists mainly have faith in and implement 3Rs approaches conveyed by their peers [11]. More openness in exchanging information between colleague researchers is considered the most critical factor for better use of 3Rs knowledge [11]. To accelerate the efforts to disseminate 3Rs approaches accordingly, transition science may be a feasible route to disclose how all stakeholders (e.g., funders, institutions that accommodate animal experiments, researchers, and legislators) can interact to achieve this [32,33]. Transitions can be seen as a development from a current regime to a new, desired, more-sustainable regime, e.g., transition science on replacement aims to understand how to govern the shift towards phasing in new animal-free methods and how to phase out animal testing in the current system. Three levels are distinguished in transition science: “The niche level” refers to the scientific–technical realm, where innovations like non-animal methods challenge the current regime. “The regime” represents the status quo or the dominant system within institutions. In the context of, e.g., animal-free safety assessment, the regime signifies the prevailing method, which is currently the use of animals for safety evaluation. This status quo is backed by regulations, industry norms, and practices. “The landscape level” represents society at large, where societal values, attitudes of regulators, economics, academia, and technical advancements shape both the regime and niche levels [32,33]. It may even be that new alternatives putting an additional workload on researchers themselves may become apparent. When scientists are asked why 3Rs approaches are not implemented, most answer lack of time, budget, and knowledge [11]. Approximately half of the respondents answered that the implementation of refinement approaches would be improved in their research if given unlimited time and budget. It is a fact that resources to develop and implement the 3Rs are limited, even though European legislation requires the implementation of the 3Rs [4]. A measure to improve this is to involve funders who support animal experiments. One may argue that allocating funding to animal experiments imposes a moral and ethical obligation on such funders to earmark funding to develop and implement the 3Rs. Institutions accommodating animal experiments should also be involved in this work. There needs to be a local focus on allocating sufficient resources and dedicated expertise within the field of the 3Rs. In addition, the education of researchers and technicians must be prioritized. Again, one may argue that institutions who support and house animal experiments bear—apart from the legal obligations—a moral responsibility to fast-forward the 3Rs. In light of this, it is worth noting that education in the 3Rs has been proven to increase awareness [23].

## 5. Limitations

We investigated the 3Rs reporting status in primary preclinical animal publications to identify whether reporting the 3Rs may be a feasible and easy way to disseminate 3Rs approaches among peers. We cannot, per se, claim that “not reporting” equals “not implementing”. This study does not clarify how well implementation is progressing. Nevertheless, several reports have conveyed difficulties in implementing the 3Rs. Also, when we scrutinized the level of detail in the reporting, we noticed that the 3Rs sometimes seemed misinterpreted. This may imply either lack of education or a lack of consulting 3Rs experts.

We decided to investigate reporting of the 3Rs, as an indicator of 3Rs awareness amongst researchers. However, the decision to investigate this item was taken after retrieving our data from Covidence and subsequently coding them in Excel. Thus, information regarding this item is not considered in our study Appendix A. Also, we looked into the year 2009, i.e., just prior to the EU directive 2010/63, and almost a decade later, into the year 2018. The situation reported here may have improved since.

Finally, one may argue that reporting of measures taken to replace animals is not expected in primary preclinical animal experimental studies. Nevertheless, our aim was to disclose 3Rs reporting per se and, thus, reporting of replacement was included. The fact that it was not reported at all highlights the 3Rs as three separate concepts that must be accounted for separately. Each R requires a unique approach and allocation of expertise, time, and money if stakeholders are expected to comply with all 3Rs and, hence, legislation.

## 6. Conclusions

Our survey has demonstrated poor reporting of 3R-related approaches in preclinical publications. Although it seems an easy dissemination route, the reporting is currently insufficient to use this approach to guide researchers toward relevant 3Rs ideas. It is a joint responsibility for researchers, publishers, funders, and institutions to adhere to the 3Rs concept and secure dissemination and implementation of the 3Rs. It calls for more vigorous enforcement of valid 3Rs information by publishers, more attention from funders (specifically those supporting animal experiments) to earmark money to 3Rs research, and more focus from institutions to improve education and interdisciplinary collaboration at institutions in order to stimulate both the development of relevant 3Rs methods and the implementation of such methods. To understand fully how to accelerate the process, transition science may be a mean towards this goal.

## Figures and Tables

**Figure 1 animals-13-03005-f001:**
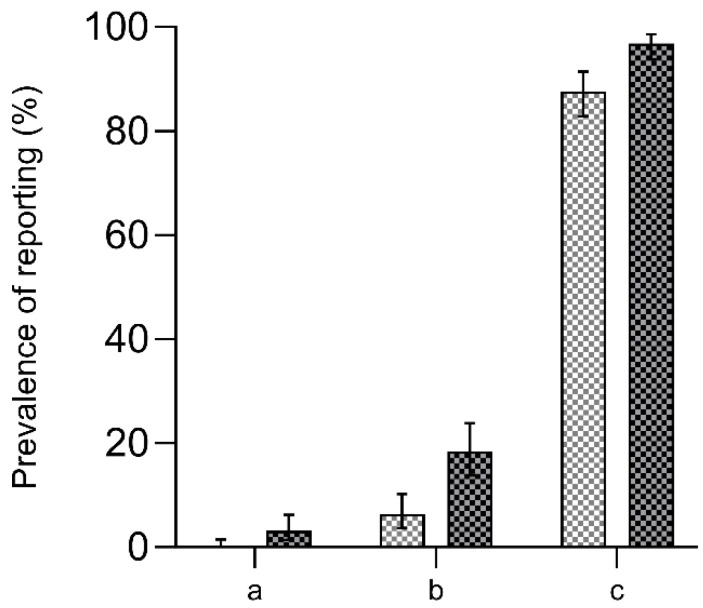
Prevalence of reporting in preclinical research in 2009 compared with 2018. Left Y-axis: prevalence of reporting in %. *x*-axis: (**a**) the 3Rs (word search for the concept); (**b**) 3Rs-related sentences of replacement, reduction, refinement; and (**c**) compliance with legislation on animal experimentation. The error bars represent 95% confidence intervals. 
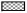
 2009, 
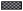
 2018.

**Figure 2 animals-13-03005-f002:**
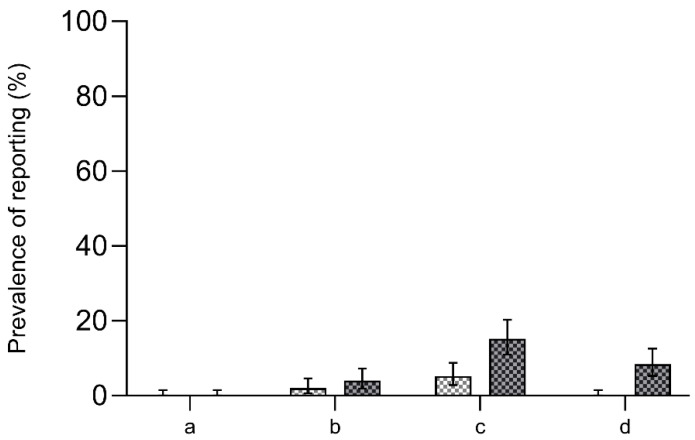
Prevalence of reporting in 2009 compared with 2018 of (**a**) replacement, (**b**) reduction, (**c**) refinement, and (**d**) compliance with the 3Rs or the ARRIVE guidelines. Left *y*-axis: prevalence of reporting in %. The error bars represent 95% confidence intervals. 
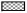
 2009 
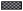
 2018.

**Table 1 animals-13-03005-t001:** Prevalence of reporting details in the preclinical research studies from 2009 and 2018.

Item	Details of Reporting	2009 (*n* = 250)			2018 (*n* = 250)			2018–2009 ^a^	
		%	95% CI	No.	%	95% CI	No.	%	95% CI
Replacement	Reported with no specific method	0.0	[0.0–1.5]	0	0.0	[0.0–1.5]	0	0.0	[0.0–0.0]
	Reported with a specific method	0.0	[0.0–1.5]	0	0.0	[0.0–1.5]	0	0.0	
Reduction	Reported with no specific method	2.0	[0.7–4.6]	5	3.6	[1.7–6.7]	9	1.6	[−1.3–4.5]
	Reported with a specific method	0.4	[0.0–2.2]	1	4.4	[2.2–7.7]	11	4.0	[1.3–6.7]
Refinement	Reported with no specific method	1.6	[0.4–4.0]	4	6.0	[3.4–9.7]	15	4.4	[1.1–7.7]
	Reported with a specific method	3.6	[1.7–6.7]	9	8.4	[5.3–13.6]	21	4.8	[0.7–8.9]
ARRIVE compliance	Reported with no checklist	NA	-	-	6.0	[3.4–9.7]	15	-	-
	Reported with a checklist	NA	-	-	1.2	[0.3–3.5]	3	-	-

CI, confidence interval; *n*, the total number of publications; No., number of publications mentioning the item; NA, non-applicable. ^a^ The relative reporting difference between 2009 and 2018.

## Data Availability

Research data are stored and available in the Open Science Framework at [DOI 10.17605/OSF.IO/2KPNM].

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
