# Peer review of "Reporting of 3Rs Approaches in Preclinical Animal Experimental Studies—A Nationwide Study"

_animals, 2023, doi:10.3390/ani13193005_

Round 1
Reviewer 1 Report
Please see attached file for comments

Some minor typos
Author Response
Dear editor and reviewer,
Thank you for all your work of reviewing our manuscript. Please find our point to point responses in the attached file.
On behalf of the authors. Kind regards,
Dr. Kousholt

Reviewer 2 Report
The authors present a systematic review and random sample of 500 preclinical studies between 2009 and 2018 to understand reporting of the 3Rs in peer-reviewed animal studies. This has high relevance to proper conduct and reporting of research using animals given the importance of the 3Rs framework in supporting ethical conduct of animal research and likelihood of researchers to look to peers for 3Rs applications and recommendations. The authors study reveals limited 3Rs information published, that is variable in quality and quantity. As a result the authors conclude that 3Rs-specific funding and education is necessary. This is an important review as it highlights the lack of norms developed around the reporting of the 3Rs despite remarkable attempts (like ARRIVE) to do so. I offer a few comments to strengthen an already well-conceived manuscript.
My comments to the authors to address are as follows:
Introduction
The authors do a good job providing concise background leading into the rationale for conducting this review. However, the statement, “To our knowledge, it is not disclosed to which extent preclinical animal experimental publications mention and promote 3Rs initiatives” is a little bit vague, and I think they can be much more straightforward in stating the gap in recent reviews evaluating the use of 3Rs practices in any given study. This is especially with the perspective that this is forming the basis for their own work, that is separate from reviews assessing uptake of the ARRIVE guidelines. Their rationale for doing so, that is scientists learn from literature existing in the field, is well understood.
Methods
Statistical approach is appropriate and I appreciated the use of ROB to manage the data/inter-rater agreement.
It was not clear to me how implementation of replacement might be captured in such an approach that selected for papers using animals in preclinical research? In vitro studies, that might detail the replacement of animals in methods, were intentionally excluded. It would be useful for the authors to describe the approach here as studies using animals with correct 3Rs implementation have determined that the model is necessary and non-animal alternatives are not valid or fit for purpose. This is somewhat addressed in 4.2 in the discussion, but it seems important to acknowledge here.
Similarly, how did the authors evaluate ‘re-use methods’, as the language that scientists might use in this situation might be highly variable, including utilizing biobanked sample or other aspects that again might not have been included in this because of the exclusion criteria.
Results
Figures 1/2 The authors include detailed legends for the figures, but the figures themselves should be more informative. Please indicate (%) on the y axis, and a/b/c is not very informative, consider altering graph orientation to be able to include more meaningful categorization on the figures themselves.
Discussion
4.1 In the discussion the authors describe low reporting of the 3Rs, but nowhere in the manuscript do they disclose the average impact factor of the journals they have surveyed or practical aspects like article length. In general high impact journals and peer reviewers discourage lengthy methods where information about the 3Rs would be found, so it is somewhat remarkable still that with this push to limit manuscript length the reporting of 3Rs is increasing, demonstrating the value placed by scientists on the 3Rs to emphasize in reporting.
4.2 Yes, the model selection is generally the focus, not models that were unselected because of poor suitability. It would be unusual to spend text in a regular field manuscript describing why you did not do perform work in an alternate model. This is unsurprising that the search methodology used by the authors did not find success in reporting Replacement, and the authors should consider their own suggestion to evaluate a subset of in vitro studies to understand whether these studies report their use as alternatives to animal experimental studies. This touched on in limitations as ‘not expected’, which is perhaps an oversimplification of the complexity in preparing a scientific manuscript intended for multiple audiences.
4.3 Fully agree with the authors that incomplete reporting impedes translation into practice, this is a highly important point. The sentence, “Sadly in terms of reporting refinement and reduction implemented in one’s study only a few scientists reported relevant details” is it the authors intent to moralize the data? if yes, this is ok. The rest of the manuscript generally seems more oriented to report findings and possibilities leading there – I wonder if this is not also a place to consider the factors that drive this behavior? Seems again important to consider the role of publishing practices if the authors suggest this is a tool to engage scientists in their field journals. High levels of detail in methods are now less common, evident also in the authors comments in the following section of reporting 4.4, where journals endorse but rarely effectively evaluate compliance and are using checklist approaches to simplify.
4.5 this is an excellent view on the interconnectedness of various disciplines, and how targeted implementation-focused funding could have a major impact. Define transition science briefly here for the reader.
In the limitations, the authors did an excellent job of touching on the challenges of a review like this. That being said, the authors were critical that sometimes 3Rs terms were misused by the authors of the manuscripts they studied, in the same way it is likely that scientists are not using terminology that is consistent with the 3Rs because of field terminology differences while they may be practicing 3Rs techniques without proper attribution.
I congratulate the authors on a highly informative manuscript that will be of great interest and relevance to researchers and supporting peers, publishers and funders to support better implementation of the 3Rs. The authors did an exceptional job of using evidence from their review to give meaning to their call for 3Rs research and training.
Author Response
Dear editor and reviewer 2,
Thank you very much for the effort you have put into the review of our manuscript. Please find our point to point responses in the attached file.
Kind regards,
Dr. Kousholt

Round 2
Reviewer 1 Report
The authors addressed all points from my first review sufficiently. Reads very nicely now. Thanks!
Author Response
We thank you very much for a thorough review